# High Expression of a Cancer Stemness-Related Gene, Chromobox 8 (CBX8), in Normal Tissue Adjacent to the Tumor (NAT) Is Associated with Poor Prognosis of Colorectal Cancer Patients

**DOI:** 10.3390/cells11111852

**Published:** 2022-06-06

**Authors:** Lui Ng, Hung-Sing Li, Abraham Tak-Ka Man, Ariel Ka-Man Chow, Dominic Chi-Chung Foo, Oswens Siu-Hung Lo, Roberta Wen-Chi Pang, Wai-Lun Law

**Affiliations:** Department of Surgery, School of Clinical Medicine, Li Ka Shing Faculty of Medicine, The University of Hong Kong, Hong Kong SAR, China; lhsing2@connect.hku.hk (H.-S.L.); tkman@hku.hk (A.T.-K.M.); ariel115@gmail.com (A.K.-M.C.); ccfoo@hku.hk (D.C.-C.F.); oswenslo@yahoo.com (O.S.-H.L.); lui_ng@hotmail.com (R.W.-C.P.)

**Keywords:** CBX8, colorectal cancer, normal tissue adjacent to the tumor (NAT), prognostic biomarker

## Abstract

Background: Several studies have demonstrated that the molecular profile of normal tissue adjacent to the tumor (NAT) is prognostic for recurrence in patients with different cancers. This study investigated the clinical significance of CBX8 gene expression, a cancer stemness-related gene, in tumor and NAT tissue of colorectal cancer (CRC) patients. Methods: The gene level of CBX8 in paired CRC and NAT specimens from 95 patients was determined by quantitative PCR. CBX8 protein level in CRC and NAT specimens from 66 patients was determined by immunohistochemistry. CBX8 gene and protein levels were correlated with the patients’ clinicopathological parameters and circulatory immune cell profiles. The association between *CBX8* and pluripotency-associated genes was analyzed using the TCGA database. Results: NAT *CBX8* gene level positively correlated with TNM stage, tumor invasion, lymph node metastasis and distant metastasis, indicating its association with tumor progression and metastasis. There was no correlation between NAT CBX8 protein level and clinicopathological parameters. Moreover, a high level of CBX8 gene and protein in NAT both correlated with poor DFS and OS. There was an inverse correlation between *CBX8* gene level and post-operative platelet counts and platelet to lymphocyte level, suggesting its association with systematic inflammation. Finally, TCGA analysis showed that *CBX8* level was correlated with a couple of pluripotency-associated genes, supporting its association with cancer stemness. Conclusions: High NAT *CBX8* is a poor prognostic factor for tumor progression and survival in CRC patients.

## 1. Introduction

Colorectal cancer (CRC) is one of the commonest cancers and a leading cause of cancer deaths worldwide. The 5-year survival rate remains below 65% in developed countries and under 50% in developing countries [1], mainly due to recurrence and metastasis. Recent studies demonstrated that the molecular profile of normal tissue adjacent to the tumor (NAT) is prognostic for recurrence in patients with different cancers including CRC [2,3,4], HCC [5], prostate [6], oral carcinoma [7], breast cancer [8] and head and neck cancer [9]. We believe a better understanding of the molecular profile in NAT is crucial for identifying molecular biomarkers for predicting the prognosis of CRC patients and therapeutic targets to combat recurrence.

Epigenetic gene repression by Chromobox (CBX) proteins is crucial for the maintenance and self-renewal capabilities of pluripotent and multipotent stem cells [10]. Chromobox 8 (CBX8) is one of the members of CBX protein family. Previous studies demonstrated that CBX8 is overexpressed in CRC and it functionally induces CRC cell proliferation and inhibits tumor apoptosis [11,12], suggesting that CBX8 plays an oncogenic role in CRC. More importantly, a recent study performed by Zhang et al. showed that CBX8 contributes to increased cancer stemness and decreased chemosensitivity in CRC [13]. In line with this study, CBX8 is also reported to induce stemness in HCC [14] and breast cancer [15]. Moreover, the prognostic potential of tumor CBX8 expression for the survival of CRC patients has been investigated in three studies [12,16,17], yet the findings were ambiguous. Tang J. et al. showed in their own clinical specimens that CBX8 protein level was overexpressed in CRC when compared to paired adjacent normal tissue, but surprisingly low expression level of CBX8 was associated with poor disease-free survival and overall survival [12]. Song X. et al. on the other hand, by analyzing an independent dataset (GSE38832) reported that a high *CBX8* gene level was significantly associated with worse overall survival [16]. In a recent study, Li Q. et al. showed no correlation between *CBX8* and disease-free survival or overall survival by using GEPIA and Kaplan–Meier plotter online database analysis tools [17]. These findings suggested that the prognostic significance of tumor CBX8 level remains inconclusive. Moreover, the significance of NAT CBX8 gene expression has not been reported. Hence, the clinical significance of CBX8 expression in tumor and NAT of CRC patients warrants further investigation for a better understanding of its prognostic significance for CRC recurrence.

This study investigated the association of tumor and NAT CBX8 level with clinicopathological parameters and disease-free and overall survival of CRC patients. The correlation between CBX8 level and CRC progression is further explained by its correlation with changes in immune cell profile and co-expression with other pluripotency-associated genes.

## 2. Materials and Methods

### 2.1. Patient Specimens

Specimens of the CRC tumor and its normal tissue adjacent to the tumor (NAT) counterpart were obtained from 95 patients who underwent surgical resection of primary CRC between 2008 and 2012 at the Department of Surgery, Queen Mary Hospital, the University of Hong Kong. Based on the guidelines of TCGA which clearly specifies that NAT samples must be collected at a distance greater than 2 cm from the tumor margins and/or the tissue must be histopathologically validated to not contain any tumor cells [18,19,20,21], all NAT specimens were collected at a distance ranging from 2 cm (minimum distance from the tumor margin) to a maximum distance of 10 cm. All samples were immediately frozen in liquid nitrogen and kept at −80 °C until analysis. The study was approved by the Institutional Review Board of the University of Hong Kong/Hospital Authority Hong Kong West Cluster (HKU/HA HKW IRB) and written informed consents were obtained from patients prior to their inclusion. Comprehensive CRC patient demographic and clinical characteristics included the year of diagnosis, age, sex, race/ethnicity, tumor location, tumor size, tumor stage, and receipt of cancer treatment (such as surgery, chemotherapy, and radiation therapy). 

### 2.2. Blood Immune Cell Profile of CRC Patients

Preoperative and postoperative immune cell count data including lymphocyte, neutrophil, neutrophil-to-lymphocyte ratio (NLR), platelet (PLT), platelet-to-lymphocyte ratio (PLR), monocyte, lymphocyte-to-monocyte ratio (LMR), albumin, and prognostic nutritional index (PNI) were obtained from the patient data database. Preoperative data were defined as results from the final blood test performed before surgery (usually one day before operation). Postoperative data were defined as those taken within 21–90 days after surgery but before starting adjuvant chemotherapy to minimize the influence of the systemic inflammatory response due to the operation and the effect of chemotherapy [22,23]. The dynamic changes in their level between post- and pre-operative blood were calculated as delta values (post-operative level minus pre-operative level).

### 2.3. RNA Extraction and Quantitative PCR (qPCR) Analysis 

Total RNA was extracted from tumors and NAT using Trizol reagent and RNA miniprep extraction kit according to the manufacturer’s instruction (Life Technologies) and 500 ng of each RNA sample was used to prepare cDNA using PrimeScript™ RT Master Mix (TaKaRa). For qPCR, 1 μL of cDNA template was added to a master mix that consists of SYBR Premix EX Taq II (TaKaRa), forward and reverse primers (5′ to 3′, *CBX8* forward: GGC CTC GGA GAG TAC CTC AA, reverse: TAG GGC TGG TCA CAC CAC AC; *GAPDH* forward: TTC ACC ACC ATG GAG AAG GC, reverse: GGC ATG GAC TGT GGT CAT GA), and ROX reference dye to form a 15 μL reaction mixture. A duplicate of each sample was made for each run. The reaction was conducted with 7900HT Fast Real-Time PCR System (Applied Biosystems, Waltham, MA, USA). The relative CBX8 expression was expressed as fold to GAPDH, which was calculated with the 2^−ΔCt(CBX8-GAPDH)^ formula.

### 2.4. Expression Correlation of CBX8 with Other Pluripotency-Associated Genes Using an Online Database

To analyze the association of CBX8 with CSC features, we analyzed its co-expression with other pluripotency gene markers in CRC and NAT from the TCGA database at GEPIA (http://gepia2.cancer-pku.cn/#index (accessed on 20 December 2021)).

### 2.5. Immunohistochemistry

Human colorectal specimens and mice xenograft tumors were first paraffin-embedded into blocks; the tissue blocks were then cut into 5 μm histosections and prepared on glass slides. The slides were deparaffinized using xylene, followed by gradual rehydration from alcohol to water. Antigen retrieval was done by heating for 10 min in pH 6.0 citrate buffer using a microwave. The slides were then immersed in 3% hydrogen peroxide and 5% horse serum in TBST to block endogenous peroxidase activity and any non-specific binding of immunoglobulin, respectively. The sections were incubated in anti-human 1:30 CBX8 (Novus Biologicals) antibody overnight at 4 °C. After 1 h incubation of biotinylated secondary antibody at room temperature, the slides were further incubated at room temperature with streptavidin-peroxidase complex for 1 h. 3,3′-diaminobenzidine chromogen solution was used to develop the signal, followed by haematoxylin for counterstaining. Dako LSABTM+ kit (Dako, Glostrup, Denmark) was used in this part of the study. All slides were assessed independently by two investigators and the extent of signal was evaluated by the staining intensity and density (score from 0 to 4). The score was the average value of the two investigators.

### 2.6. Statistical Analysis

Student’s *t*-test was used to analyze differences between experimental groups of clinical specimens. Survival curves were generated using the Kaplan–Meier test and compared by the log-rank test. The association between CBX8 expression and immune cell profile was analyzed with Pearson and Spearman correlation. The criterion for statistical significance was *p* < 0.05. All statistical analyses were conducted using SigmaPlot version 10.0 (Systat Software Inc., San Jose, CA, USA).

## 3. Results

### 3.1. CBX8 Gene and Protein Expressions in CRC and Normal Tissue Adjacent to Tumor (NAT) Specimens and Their Clinicopathological Significances

The expression of *CBX8* in 95 paired human CRC and adjacent non-tumor specimens was determined by quantitative PCR and correlated with the patients’ clinicopathological data. The expression of *CBX8* was significantly higher in CRC compared to the NAT specimens (Figure 1A, *p* < 0.001). On average, the *CBX8* level in CRC was ~2.92-fold higher than NAT (Figure 1B). More specifically, 55 out of the 95 patients demonstrated at least 2-fold *CBX8* overexpression in CRC. The correlation of *CBX8* levels in CRC and NAT speicmens with patients’ clinicopathological data was subsequently analyzed (Table 1). We found that tumor *CBX8* level was not associated with any clinical parameters, except a trend of inverse correlation with tumor invasion was seen (*p* = 0.152). This observation is in line with a previous study showing high *CBX8* impaired CRC metastasis [12]. On the other hand, high NAT *CBX8* showed a positive correlation with stage (*p* = 0.026), tumor invasion (*p* = 0.0035), lymph node metastasis (*p* = 0.036) and distant metastasis (*p* = 0.044), indicating its association with tumor progression and metastasis.

Among these 95 patients, formalin-fixed and parafilm-embedded blocks were available from 66 of them, hence we performed immunohistochemistry to determine the protein level of CBX8 in these patients. CBX8 was significantly overexpressed in CRC when compared to paired NAT (*p* < 0.001, Figure 1B,C). In CBX8-positive NAT specimens, both crypt epithelial cells and stroma cells were able to express CBX8 (Appendix A). The CBX protein level was significantly correlated with CBX8 gene level in NAT (R = 0.405, *p* = 0.00074) but not in CRC (Figure 1D). No significant correlation was found between CRC nor NAT CBX8 protein level and clinicopathological parameters.

### 3.2. Prognostic Significance of CBX8

We investigated the association of CBX8 with 5-year disease-free survival (DFS) and overall survival (OS) of CRC patients. Our data showed that the tumor *CBX* gene level was not associated with DFS nor OS of patients (data not shown). On the other hand, when we stratified the patients into high NAT *CBX8* (relative level above 0.000288, i.e., the upper quartile level) or low NAT *CBX8* (relative level below 0.000166, i.e., the lower quartile level), patients with high NAT *CBX8* demonstrated significantly worse DFS (1224 vs. 1618 days, *p* = 0.016) and OS (1223 vs. 1651 days, *p* = 0.023), compared to those with low NAT *CBX8* level (Figure 2A,B).

We also investigated the correlation between CBX8 protein levels and prognosis of the patients (Figure 2C–F). Our data showed that high NAT CBX8 protein levels (IHC score ≥ 2.5, i.e., the upper quartile level), when compared to low NAT CBX level group (IHC score ≤ 1.5, i.e., the lower quartile level), showed significantly shorter DFS (1081 vs. 1478 days, *p* = 0.020) and OS (1195 vs. 1559 days, *p* = 0.041). Similarly, patients with high CRC CBX8 protein levels (IHC score ≥ 3.5, i.e., the upper quartile level) also demonstrated significantly shorter DFS (1002 vs. 1582 days, *p* = 0.042) and OS (1150 vs. 1658 days, *p* = 0.036) when compared to the low CRC CBX group (IHC score ≤ 1.5, i.e., the lower quartile level). 

To summarize, our clinical analysis demonstrated that high CBX8 gene and protein levels in NAT or CRC are associated with poor prognosis in CRC patients. Furthermore, the NAT *CBX8* gene level is closely associated with clinicopathological parameters related to tumor progression and metastasis.

### 3.3. Association between CBX8 Expression and Immune Cell Profile of CRC Patients

The immune system plays the most vital role in protection against diseases, including cancer. We previously reported that certain immune cell levels are correlated with the prognosis of post-operative CRC patients [24]. Hence, in this study, we investigated the association between tumor and NAT CBX8 level with immune cell profile of pre-operative and post-operative blood samples, as well as the dynamic changes in their level between post- and pre-operative blood (expressed as a delta value, which is calculated as post-operative level minus pre-operative level). 

Our results showed that there was no correlation between tumor nor NAT CBX8 level with any pre-operative blood parameters. On the other hand, NAT *CBX8* gene level was inversely associated with post-operative PLT (R = −0.342, *p* = 0.0008, Figure 3A) and PLR (R = −0.333, *p* = 0.01, Figure 3B), whereas low PLT and PLR were associated with inferior OS in patients with metastatic CRC [25]. Moreover, NAT *CBX8* gene level showed a positive trend with post-operative CEA (R = 0.233, *p* = 0.075, Figure 3C), which is a risk factor for recurrence [26,27,28,29]. On the other hand, tumor *CBX8* gene level was associated with post-operative LMR (R = 0.479, *p* = 0.0001, Figure 3D), which indicated a better prognosis [23]. In addition, a higher tumor *CBX8* gene level was correlated with lower delta-LMR (R = −0.433, *p* = 0.0007, Figure 3E) and higher delta-eosinophil (R = 0.302, *p* = 0.0201, Figure 3F). Though the prognostic value of these parameters has not been reported in CRC, low delta-LMR and high delta-eosinophil were associated with better OS or DFS in renal cell carcinoma and head and neck cancer patients [30,31], respectively. These correlations suggested that tumor *CBX8* gene level is associated with better prognosis in CRC patients.

Comparing the *CBX8* gene levels, less correlation was found between CBX8 protein level in NAT or CRC with immune cell profile of the patients. Surprisingly, CRC CBX8 protein level demonstrated an inverse correlation with delta-eosinophil, which is opposite to the pattern observed for CRC CBX8 gene level (R = −0.361, *p* = 0.0281, Figure 4A). Moreover, CBX8 protein levels in both NAT and CRC were positively correlated with post-operative CEA levels and delta-CEA levels (Figure 4B–E). These findings indicate that high NAT and CRC CBX8 levels are poor prognostic factors for CRC patients.

### 3.4. Association of CBX8 with Pluripotency-Associated Genes in CRC and NAT

CBX8 has been shown to induce cancer stemness and chemoresistance in CRC through activation of the transcription of LGR5 in a noncanonical manner [13]. To further demonstrate its association with CSC features, we analyzed its co-expression with other pluripotentcy gene markers in CRC from the TCGA database at GEPIA (http://gepia2.cancer-pku.cn/#index (accessed on 21 February 2022)). In line with the above study, *CBX8* was positively correlated with LGR5 in CRC (Table 2). Furthermore, we found that *CBX8* level in colon cancer was positively correlated with a couple of pluripotentcy-associated genes including *BMI1*, *OCT4*, *MYC*, *USP36*, *RPTOR*, *UBE2O*, *JMJD6*, *SLCO4A1*, *UTP18*, *E2F6* and *CD44*. These data demonstrated the close association between *CBX8* and colorectal cancer stemness.

We are also interested in the correlation between *CBX8* and pluripotency-associated genes in NAT of colorectal cancer patients (Table 2). Since the sample number is lower for NAT at GEPIA, we considered pluripotency-associated genes, which showed a correlation with *CBX8* (*p*-value below 0.1), as noteworthy candidates. Similar to the finding in CRC specimens, the significant correlations of *CBX8* with *MYC* and *UBE2O* were also observed in NAT, and a trend of correlations were also observed for *LGR5*, *UTP18* and *ABHD5*. Furthermore, a trend of positive correlation with CD44 was also shown in NAT but not in CRC. These results suggested that high *CBX8* level in NAT might indicate the presence of CSC-like cells within NAT that lead to recurrence and hence poor DFS and OS.

## 4. Discussion

Previous studies demonstrated that CBX8 is overexpressed in CRC and it functionally induces CRC cell proliferation and inhibits tumor apoptosis [11,12], and CBX8 is associated with cancer stemness and chemoresistance in CRC [13], yet the prognostic potential of tumor CBX8 expression for the survival of CRC patients was ambiguous among current reports [12,16,17]. Hence this study comprehensively investigated the clinicopathological and prognostic significance of CBX8 gene and protein levels in both NAT and CRC specimens.

This study demonstrated that high NAT CBX8 gene and protein level was associated with higher risk of recurrence and mortality. The significance of NAT tissue has been suggested since Slaughter et al. first described the “field cancerization” theory [32], suggesting a cumulative process of carcinogenesis in which genetic alterations are acquired step-wise, leaving the NAT tissue in an intermediate, pre-neoplastic state composed of morphologically normal but molecularly altered cells. A recent study on histologically normal breast epithelium by Graham K et al. suggested that the microenvironment surrounding the tumor is essential for understanding recurrence [33]. In line with this phenomenon, the NAT molecular profile is reported to show the prognostic significance for recurrence in patients with different cancers including CRC [2,3,4], HCC [5], prostate [6], oral carcinoma [7], breast cancer [8] and head and neck cancer [9]. Further investigation on CBX8, as well as other dysregulated NAT molecular markers, for prognosis of CRC patients, warrants evaluation for their clinical use. 

On the other hand, the tumor CBX8 gene and protein level demonstrated a rather opposite pattern in its prognostic significance. Although tumor *CBX8* gene level was not associated with DFS and OS of CRC patients in this study, our findings demonstrated that higher *CBX8* level in tumor was associated with higher post-operative LMR and lower delta LMR, which have been reported as factors of better prognosis for CRC patients [30,31]. In addition, our data revealed that higher tumor *CBX8* showed a trend towards lower tumor invasion. On the other hand, high CRC CBX8 protein level was associated with poor survival and higher CEA. Since our data showed that there was no significant correlation between CBX8 gene and protein level, this might explain the discrepancies observed in our study as well as among the different CBX8 studies within CRC patients.

Chromobox protein homolog 8 (CBX8), together with CBX2, CBX4, CBX6, and CBX7, forms the CBX family, which is the central part of the polycomb repressive complex 1 (PRC1) [34]. CBX8 has recently emerged as a potential oncogenic target in multiple malignancies [35]. Previous studies demonstrated that CBX8 is overexpressed in CRC and it functionally induces CRC cell proliferation and inhibits tumor apoptosis [11,12], suggesting that CBX8 plays an oncogenic role in CRC. More importantly, CBX8 contributes to increased cancer stemness in CRC [13], as well as HCC [14] and breast cancer [15]. This study demonstrated the association between *CBX8* level and a couple of pluripotency-associated genes including *LGR5*, *Bmi1*, *OCT4*, *MYC*, *USP36*, *RPTOR*, *UBE2O*, *JMJD6*, *SLCO4A1*, *UTP18* and *E2F6*, and inversely with *CD166*, *ABHD5*, *PDCD4* and *GPA33*. Of note, CD166 is associated with a decreased risk of vascular invasion in CRC in a meta-analysis consisting of 3332 cases [36]. In addition, ABDH5 impairs colon cancer stemness [37]; PDCD4 is a well-known tumor suppressor in CRC [38] and it represses stemness of adipose-derived stem cells [39], gastric [40] and cervical cancer cells [41]; GPA33 is a cell surface antigen expressed across a panel of CSC derived from various stages of CRC, in addition to exhibiting 100% penetrance across over 50 primary and metastatic colorectal cancer specimens, and its application as a therapeutic target is in progress [42,43]. These findings indicate the high *CBX8* expression in CRC is indeed related to its CSC feature. *CBX8* was also correlated with *MYC*, *UBE2O*, *UTP18*, *ABHD5*, *LGR5* and *CD44* in NAT, suggesting that a high *CBX8* level in NAT might indicate the presence of CSC-like cells within NAT that can lead to recurrence.

In fact, CSCs are known to be responsible for therapeutic resistance and c-MYC plays a pivotal role in the regulation of CSCs [44]. c-MYC is upregulated in several cancers and the TCGA database shows that 28% of all cancers harbour genetic abnormalities in at least one of the Myc family [45]. c-MYC contributes to tumor progression and metastasis and induces cancer stemness properties in multiple neoplasms [46]. It also regulates the transcription of over 15% of human genes, such as *CCND*, *CDK4* and *E2F1*, which are involved in tumour progression [47]. Furthermore, Strippoli et al. showed that alterations in the transcriptional factor c-MYC are involved in anti-EGFR resistance in metastatic colorectal cancer; patients with higher c-MYC expression showed a significantly lower PFS and OS when compared to those with low c-MYC expression [48]. Moreover, with expression gene profiling, they pointed out the pivotal role of c-MYC in CRC-related cell-cycle, apoptosis, signal transduction and cell-growth pathways. More importantly, our supplementary experiments showed that CBX8 overexpression in the CRC cell line HCT116 induced the protein level of c-MYC (Appendix A), as well as repressed the level of *let-7*, which is reported to downregulate c-MYC in CRC as well as being down-regulated by c-MYC via LIN28 repression [49]. These results suggest that CBX8 not only was associated with the level of CSC-associated genes, but also functionally regulated the CSC features through regulating c-MYC and possibly other CSC-related genes. It is also reported that UBE2C, a gene that encodes an enzyme required for cell cycle progression, is directly involved in cetuximab resistance and could be a valid target to overcome the EGFR-blockage resistance [50]. A gene belonging to the same family (UBE2O) has been shown to promote the proliferation, epithelial–mesenchymal transformation and stemness properties of breast cancer cells through the UBE2O/AMPKα2/mTORC1-MYC positive feedback loop [51]. This same gene has been demonstrated to be correlated with *CBX8* levels both in colon cancer and in NAT, emphasizing the intricate interconnection between *CBX8* levels, stemness, tumor progression and resistance to therapy.

In light of the functional significance of CBX8 in inducing malignancy in multiple cancer types, it has been suggested as a therapeutic target for cancer treatment and very recently its inhibitors have been developed and are being tested [35,52,53]. Our study showed that patients with high NAT *CBX8* were associated with a worse stage, tumor invasion, lymph node metastasis, distant metastasis, and poor DFS and OS, suggesting that CRC patients with high NAT *CBX8* are potential candidates for CBX8-targeted treatment. Moreover, we showed that post-operative PLT and PLR levels were associated with NAT *CBX8* levels. It is warranted to investigate the potential of PLR and PLT as biomarkers to predict and monitor the response to CBX8 inhibitor treatment.

The limitation of this study is that the functional significance of CBX8 in NAT was not investigated. Nevertheless, CBX8 has been demonstrated to induce tumor growth or progression in CRC and other cancers, and our supplementary experiment indicated that its overexpression induced the protein level of c-MYC. We believe the high expression of CBX8 in NAT is not just a biomarker but is one of the factors contributing to tumor progression, recurrence and metastasis. Therefore, further investigation of the treatment effect on recurrence by repressing its expression in NAT in animal models following primary tumor removal, is warranted to evaluate the potential of treating post-operative CRC patients using this novel approach. 

## Figures and Tables

**Figure 1 cells-11-01852-f001:**
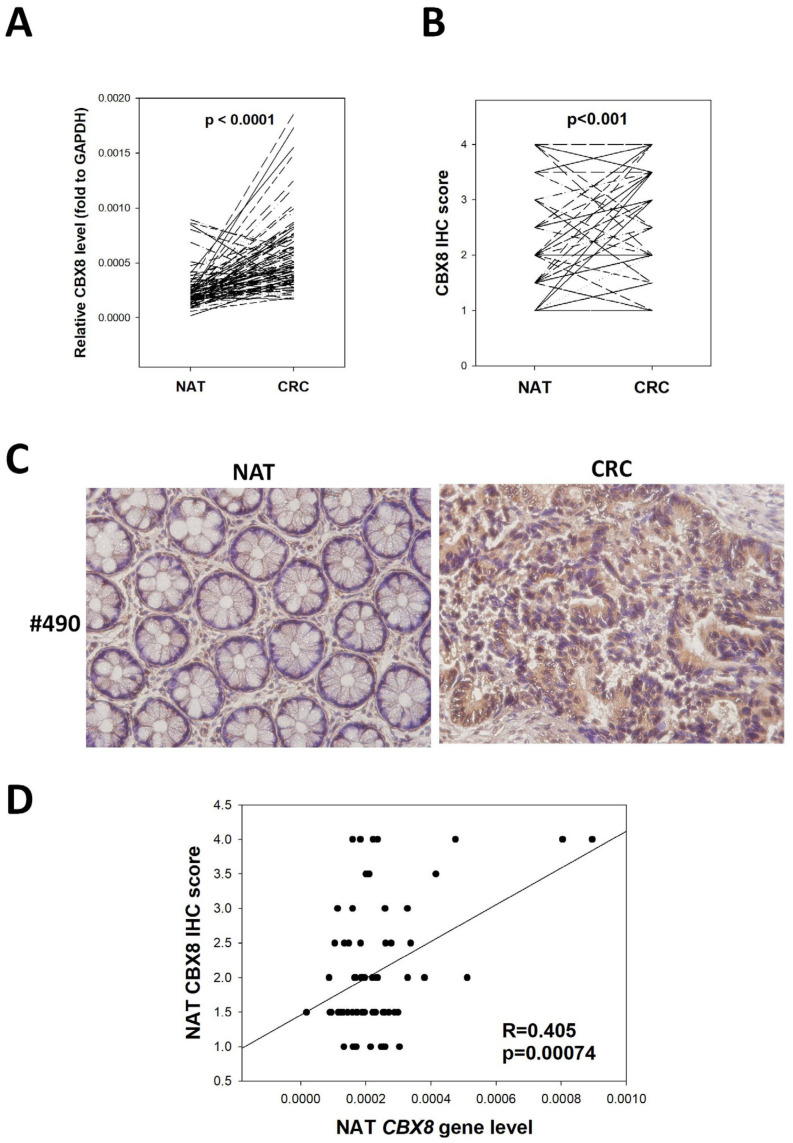
Clinical significance of CBX8 expression in CRC patients. (**A**) The *CBX8* gene was significantly overexpressed in CRC when compared to paired NAT tissue (*n* = 95). (**B**) The CBX8 protein level was significantly higher in CRC when compared to paired NAT tissue (*n* = 66). (**C**) Representative IHC stainings of CBX8 in patient #490 showed higher CBX8 level in CRC compared to paired NAT (Magnification: 40×). (**D**) A significant correlation was observed between CBX8 gene and protein levels within the NAT tissue (*n* = 66).

**Figure 2 cells-11-01852-f002:**
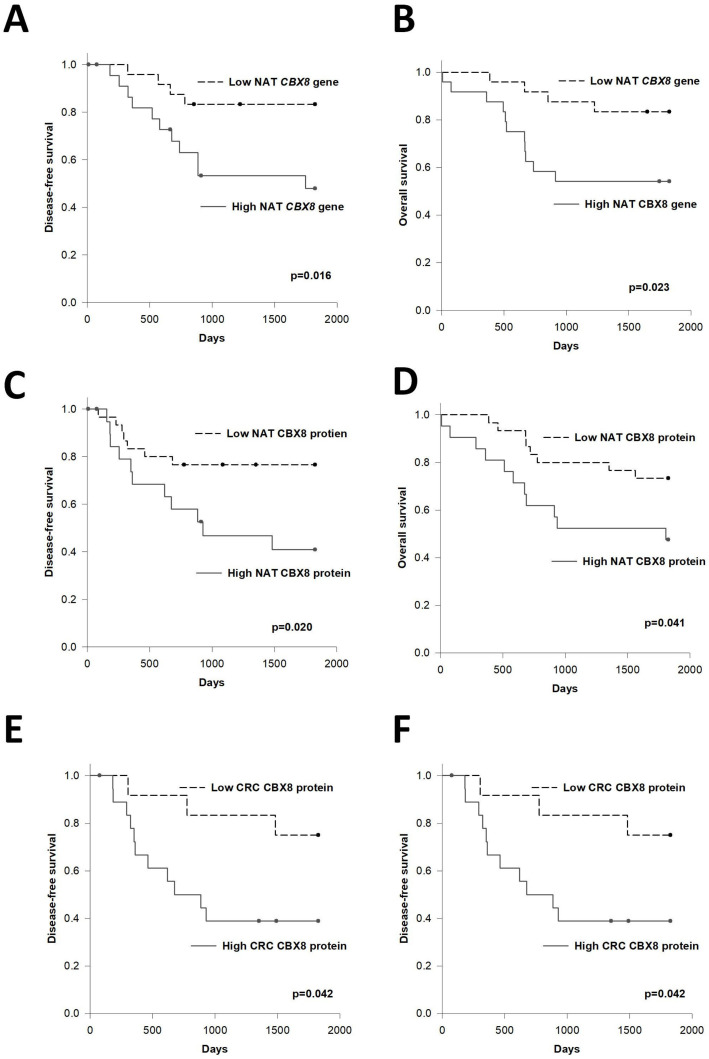
Correlation between CBX8 level and survival of CRC patients. (**A**,**B**) CRC patients with high NAT *CBX8* gene levels, compared to those with low NAT *CBX8* levels, have a significantly poorer (**A**) disease-free survival and (**B**) overall survival. (**C**,**D**) CRC patients with high NAT CBX8 protein levels, compared to those with low NAT CBX8 levels, have a significantly poorer (**C**) disease-free survival and (**D**) overall survival. (**E**,**F**) CRC patients with high CRC CBX8 protein levels, compared to those with low CRC CBX8 levels, have a significantly poorer (**E**) disease-free survival and (**F**) overall survival.

**Figure 3 cells-11-01852-f003:**
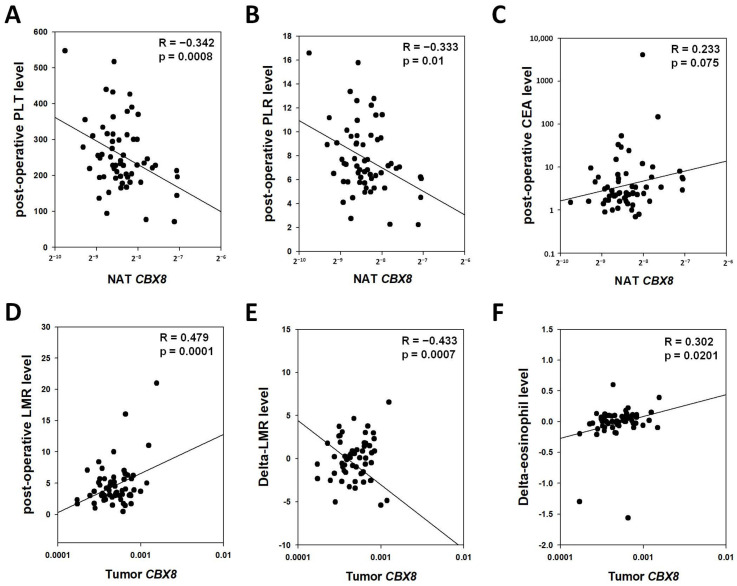
Correlation between NAT or CRC *CBX8* gene expression and blood immune cell profiles in CRC patients. (**A**,**B**) NAT *CBX8* level was inversely correlated with level of (**A**) platelets (PLT) and (**B**) platelet to lymphocyte ratio (PLR) in post-operative blood of CRC patients. (**C**) NAT *CBX8* level showed a positive trend with level of post-operative CEA. (**D**) Tumor *CBX8* level was positively correlated with post-operative LMR level. (**E**,**F**) Tumor *CBX8* level was inversely and positively correlated with dymamic changes (post-operative minus pre-operative) in (**E**) LMR level and (**F**) eosinophil level, respectively.

**Figure 4 cells-11-01852-f004:**
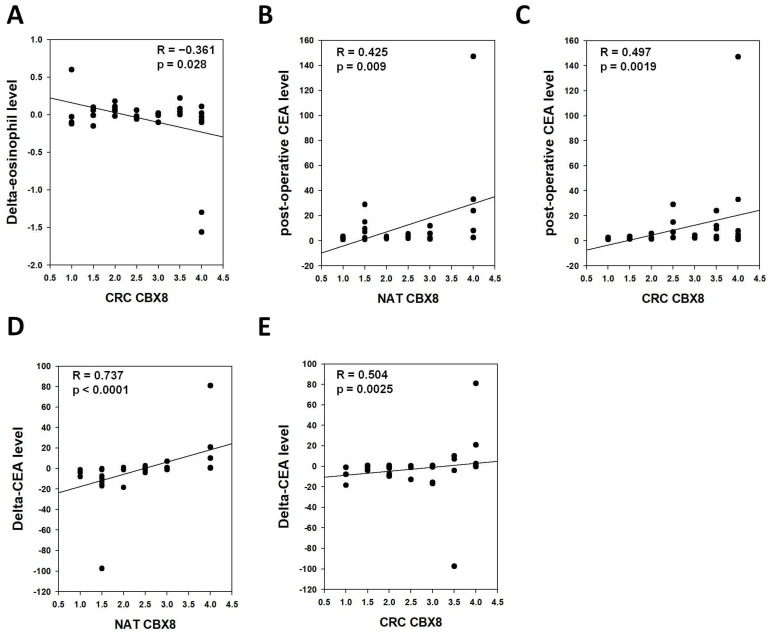
Correlation between NAT or CRC CBX8 protein level and blood immune cell profiles in CRC patients. (**A**) The CRC CBX8 level was inversely correlated with dyanmic changes (post-operative minus pre-operative) in eosinophil level. (**B**,**C**) CBX8 protein levels in (**B**) NAT and (**C**) CRC were positively correlated with post-operative CEA level. (**D**,**E**) CBX8 protein levels in (**D**) NAT and (**E**) CRC were positively correlated with dynamic changes (post-operative minus pre-operative) in CEA level.

**Table 1 cells-11-01852-t001:** Clinicopathological correlation of CBX8 level in Tumor and NAT specimens of 95 CRC patients.

Parameter	Category	Number of Cases	Tumor CBX8	*p*-Value	NAT CBX8	*p*-Value
Gender	Male	52	0.000592	NS	0.000251	NS
	Female	43	0.000526		0.000260	
Age	≤70	36	0.000543	NS	0.000249	NS
	>70	59	0.000564		0.000260	
Location	Colon	63	0.000575	NS	0.000292	0.004
	Rectum	32	0.000520		0.000184	
Tumor Size	<5 cm	54	0.000546	NS	0.000237	NS
	≥5 cm	41	0.000569		0.000281	
TNM Staging	I to II	39	0.000584	NS	0.000208	0.026
	III to IV	56	0.000537		0.000289	
Tumor invasion	T1 to T3	79	0.000578	NS	0.000239	0.035
	T4	16	0.000449		0.000339	
Lymph node metastasis	Absent	48	0.000544	NS	0.000219	0.034
	Present	47	0.000569		0.000294	
Distant Metastasis	Present	73	0.000572	NS	0.000236	0.044
	Absent	22	0.000506		0.000321	

NS: not significant (*p* > 0.05).

**Table 2 cells-11-01852-t002:** Correlation between expression of *CBX8* and pluripotency-associated genes in CRC and NAT specimens from the TCGA database.

	Tumor		NAT	
	Correlation Coefficient (R)	*p*-Value	Correlation Coefficient (R)	*p*-Value
Positive correlation				
*LGR5*	0.12	0.02	0.24	0.095 *
*BMI1*	0.13	0.031		NS
*OCT4*	0.16	0.0017		NS
*MYC*	0.23	<0.0001	0.3	0.0063
*USP36*	0.46	<0.0001		NS
*RPTOR*	0.41	<0.0001		NS
*UBE2O*	0.49	<0.0001	0.22	0.045
*JMJD6*	0.44	<0.0001		NS
*SLCO4A1*	0.38	<0.0001		NS
*UTP18*	0.42	<0.0001	0.28	0.08 *
*E2F6*	0.42	<0.0001		NS
*CD44*		NS	0.29	0.062 *
Inverse correlation				
*CD166*	−0.13	0.01		NS
*ABHD5*	−0.21	0.00034	−0.3	0.053 *
*PDCD4*	−0.24	<0.0001		NS
*GPA33*	−0.28	<0.0001		NS

NS: not significant. * denotes trend of correlation that the *p*-value is below 0.1.

## Data Availability

The data presented in this study are available on request from the corresponding author. The data are not publicly available due to privacy.

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
