# Peer review of "High Expression of a Cancer Stemness-Related Gene, Chromobox 8 (CBX8), in Normal Tissue Adjacent to the Tumor (NAT) Is Associated with Poor Prognosis of Colorectal Cancer Patients"

_cells, 2022, doi:10.3390/cells11111852_

Round 1

Reviewer 1 Report

In this manuscript, Lui Ng and collaborators investigated the clinical significance of CBX8 gene expression in tumor and NAT tissue of CRC patients. Using paired CRC and NAT specimens from 95 patients, the gene level of CBX8 was determined by quantitative PCR and correlated with the patients’ clinicopathological parameters and circulatory immune cell profiles. In addition, the authors analyzed the association between CBX8 and pluripotency-associated genes by using TCGA database. They found that NAT CBX8 level positively correlated with CRC TNM stage, tumor invasion, lymph node metastasis and distant metastasis, indicating its association with tumor progression and metastasis. Moreover, high NAT CBX8 correlated with poor patients’ survival. Moreover, CBX8 level was correlated with a couple of pluripotency-associated genes, on the base of TCGA database analysis.

Overall, this manuscript is potentially of interest. However, the data presented are too preliminary and essentially based on in silico analysis. The following comments/questions should be addressed:

  • In support of their analysis, the authors should demonstrate the expression of CBX8 in NAT and CRC sections by IHC.
  • which cells in NAT are able to express CBX8?
  • Is CBX8 required for c-Myc expression or phosphorylation?

  In addition, the functional significance of CBX8 in CRC remains to be investigated.

Author Response

Response to Reviewer 1 Comments 

Overall, this manuscript is potentially of interest. However, the data presented are too preliminary and essentially based on in silico analysis. The following comments/questions should be addressed:

Point 1: In support of their analysis, the authors should demonstrate the expression of CBX8 in NAT and CRC sections by IHC.

Response 1: We added the supplementary figure 1 to demonstrate NAT and CRC specimens with high CBX8 (upper panel) and low CBX8 (lower panel) expression in two representative patients (#460 and #361, respectively). IHC was performed to determine the protein level of CBX8 in 66 patients. CBX8 was significantly overexpressed in CRC when compared to paired NAT (p<0.001, Figures 1B and 1C). The CBX protein level was significantly correlated with CBX8 gene-level in NAT (R=0.405, p=0.00074) but not in CRC (Figure 1D). Unlike CBX8 gene level in CRC or NAT which showed significant correlation with certain clinicopathological parameters, no significant correlation was found between CRC nor NAT CBX8 protein level and those parameters. But similar to the results of high NAT CBX8 gene level, high NAT CBX8 protein levels also showed poor DFS and OS. We revised the abstract section to include the findings related to CBX8 protein level in NAT.

Point 2: Which cells in NAT are able to express CBX8?

Response 2: We added the supplementary figure 1 to demonstrate NAT and CRC specimens with high CBX8 (upper panel) and low CBX8 (lower panel) expression in two representative patients (#460 and #361, respectively). As shown by sample #460 in supplementary figure 1, crypt epithelial cells and stromal cells in NAT specimen are able to express CBX8.

Point 3: Is CBX8 required for c-Myc expression or phosphorylation?

Response 3: We performed supplementary experiments and added supplementary figure 2 in the revised manuscript for responding to this question. Overexpression of CBX8 in CRC cell-line HCT116 was able to increase the expression of c-Myc. Furthermore, CBX8 overexpression repressed the gene level of let-7.MicroRNA let-7 is reported to repress c-Myc expression, whereas c-Myc is also demonstrated to transactivate Lin28B, which inhibits let-7 expression. These results suggest that CBX8 functionally regulates the c-Myc expression. We added the following paragraph in the discussion section of the revised manuscript:

“More importantly, our supplementary experiments showed that CBX8 overexpression in CRC cell-line HCT116 induced the protein level of c-MYC (supplementary figure 2), as well as repressed the level of let-7 which is reported to downregulate c-MYC in CRC as well as being down-regulated by c-MYC via LIN28 repression [49]. These results suggest that CBX8 not only was associated with the level of CSC-associated genes, but also functionally regulated the CSC features.”

Point 4: In addition, the functional significance of CBX8 in CRC remains to be investigated.

Response 4: Yes, we agreed that the limitation of this study is that the functional significance of CBX8 in NAT is not investigated. There are several studies investigated the functional significance of CBX8 in CRC. Previous studies demonstrated that CBX8 is overexpressed in CRC and it functionally induces CRC cell proliferation and inhibits tumor apoptosis. A recent study performed by Zhang et al showed that CBX8 contributes to increased cancer stemness and decreased chemosensitivity in CRC. As this study mainly focused on the clinical significance and prognostic role of CBX8 in CRC, we didn’t investigate its functional roles.

However, as concerned by the reviewers, to show that CBX8 not only was associated with the level of CSC-associated genes, but also functionally regulated the CSC features, we performed supplementary experiments and added supplementary figure 2 in this revised manuscript. Our supplementary experiments showed that CBX8 overexpression in CRC cell-line HCT116 induced the protein level of c-MYC, which is well-known for its roles in regulating CSC feature. We further showed that CBX8 overexpression down-regulated let-7 expression, which is reported to downregulate c-MYC in CRC as well as being down-regulated by c-MYC via LIN28 repression. These results suggest that CBX8 was able to induce CSC feature through regulating c-MYC and possibly other CSC-related genes.

Reviewer 2 Report

Overall, this is an interesting and timely study that investigates the role of CBX8 gene expression, a cancer stemness-related gene, in tumor and normal tissue adjacent to the tumor (NAT) in colorectal cancer. The authors showed that high NAT CBX8 expression was associated with stage, tumor invasion, lymph node metastasis, distant metastasis, and poor DFS and OS, suggesting that CRC patients with high NAT CBX8 are potential candidates who could benefit from CBX8-targeted treatment. In fact, in light of the functional significance of CBX8 in inducing malignancy in multiple cancer types, it has been suggested as a therapeutic target for cancer treatment and very recently its inhibitors are developed and being tested. The authors also investigated the association between tumor and NAT CBX8 level with immune cell profile of pre-operative and post-operative blood samples, as well as the dynamic changes of their level between post- and pre-operative blood samples: they showed that there was an inverse correlation between CBX8 and post-operative platelet counts and platelet to lymphocyte levels, suggesting its association with systemic inflammation. Finally, The Cancer Genome Atlas (TCGA) analysis showed that CBX8 level was correlated with a couple of pluripotency-associated genes, supporting its association with cancer stemness.

I think that overall, this is an accurate study on an adequate number of patients and that provides significant data on a potential therapeutic target in CRC. Nevertheless, I would like the authors to discuss in more detail the functional role of CBX8 in NAT by linking it to the concept of stemness. In fact, the authors report how CBX8 has a role in the maintenance and self-renewal capabilities of pluripotent and multipotent stem cells and they also suggest that high CBX8 level in NAT might indicate the presence of cancer stem cell (CSC)-like cells within NAT, leading to recurrence and hence poor DFS and OS. To come to this conclusion, the authors have analyzed the co-expression of CBX8 with other pluripotency gene markers in CRC from TCGA database at GEPIA. They found that CBX8 level was positively correlated with pluripotency-associated genes not only in colon cancer, but also in NAT. Similar to the finding in CRC specimens, the significant correlations of CBX8 with MYC and UBE2O were also observed in NAT, and a trend of correlations was also observed for UTP18 and ABHD5. I think the authors at this point should emphasize the role of c-MYC in cancer stem cell-related signaling and resistance to cancer chemotherapy: in fact, CSCs are known to be responsible for the therapeutic resistance and c-MYC plays a pivotal role in the regulation of CSCs [Elbadawy M, Usui T, Yamawaki H, Sasaki K. Emerging Roles of C-Myc in Cancer Stem Cell-Related Signaling and Resistance to Cancer Chemotherapy: A Potential Therapeutic Target Against Colorectal Cancer. Int J Mol Sci. 2019 May 11;20(9):2340. doi: 10.3390/ijms20092340. PMID: 31083525; PMCID: PMC6539579]. c-MYC is upregulated in several cancers and TCGA database shows that 28% of all cancers harbour genetic abnormalities in at least one of the Myc family [Schaub, F.X.; Dhankani, V.; Berger, A.C.; Trivedi, M.; Richardson, A.B.; Shaw, R.; Zhao, W.; Zhang, X.; Ventura, A.; Liu, Y.; et al. Pan-cancer alterations of the myc oncogene and its proximal network across the cancer genome atlas. Cell Syst. 2018, 6, 282–300]. c-MYC contributes to tumor progression and metastasis and induces cancer stemness properties in multiple neoplasms [Blancato, J., Singh, B., Liu, A., Liao, D. J. & Dickson, R. B. Correlation of amplification and overexpression of the c-myc oncogene in high-grade breast cancer: FISH, in situ hybridisation and immunohistochemical analyses. Br. J. Cancer 90, 1612–1619 (2004)]. It also regulates the transcription of over 15% of human genes, such as CCND, CDK4 and E2F1, which are involved in tumour progression [Meyer, N. & Penn, L. Z. Reflecting on 25 years with MYC. Nat. Rev. Cancer 8, 976–990 (2008)].  Furthermore, Strippoli et al. [Strippoli A, Cocomazzi A, Basso M, Cenci T, Ricci R, Pierconti F, Cassano A, Fiorentino V, Barone C, Bria E, Ricci-Vitiani L, Tortora G, Larocca LM, Martini M. c-MYC Expression Is a Possible Keystone in the Colorectal Cancer Resistance to EGFR Inhibitors. Cancers (Basel). 2020 Mar 10;12(3):638. doi: 10.3390/cancers12030638. PMID: 32164324; PMCID: PMC7139615] showed that alterations in the transcriptional factor c-MYC are involved in the anti-EGFR resistance in metastatic colorectal cancer: patients with higher c-MYC expression showed a significant lower PFS and OS when compared to those with low c-MYC expression. Moreover, with expression gene profiling, they pointed out the pivotal role of c-MYC in CRC-related cell-cycle, apoptosis, signal transduction and cell-growth pathways. Regarding the cell growth and proliferation pathways, they showed how UBE2C, a gene that encodes an enzyme required for cell cycle progression, is directly involved in cetuximab resistance and could be a valid target to overcome the EGFR-blockage resistance. A gene belonging to the same family (UBE2O) has been shown to promote the proliferation, epithelial–mesenchymal transformation and stemness properties of breast cancer cells through the UBE2O/AMPKα2/mTORC1-MYC positive feedback loop [Liu, X., Ma, F., Liu, C. et al. UBE2O promotes the proliferation, EMT and stemness properties of breast cancer cells through the UBE2O/AMPKα2/mTORC1-MYC positive feedback loop. Cell Death Dis 11, 10 (2020). https://doi.org/10.1038/s41419-019-2194-9]. This same gene has been demonstrated to be correlated with CBX8 levels both in colon cancer and in NAT, emphasizing the intricate interconnection between CBX8 levels, stemness, tumor progression and resistance to therapy. I think it would be useful to cite such works in order to overcome, even if only partially, the limitation linked to the absence of investigation of the functional significance of CBX8 in NAT. However, the addition of these references would guarantee a broader analysis of the topic, giving a more solid theoretical justification to the results obtained.

Author Response

Response to Reviewer 2 Comments 

I think that overall, this is an accurate study on an adequate number of patients and that provides significant data on a potential therapeutic target in CRC. Nevertheless, I would like the authors to discuss in more detail the functional role of CBX8 in NAT by linking it to the concept of stemness. In fact, the authors report how CBX8 has a role in the maintenance and self-renewal capabilities of pluripotent and multipotent stem cells and they also suggest that high CBX8 level in NAT might indicate the presence of cancer stem cell (CSC)-like cells within NAT, leading to recurrence and hence poor DFS and OS. To come to this conclusion, the authors have analyzed the co-expression of CBX8 with other pluripotency gene markers in CRC from TCGA database at GEPIA. They found that CBX8 level was positively correlated with pluripotency-associated genes not only in colon cancer, but also in NAT. Similar to the finding in CRC specimens, the significant correlations of CBX8 with MYC and UBE2O were also observed in NAT, and a trend of correlations was also observed for UTP18 and ABHD5.

Point 1: I think the authors at this point should emphasize the role of c-MYC in cancer stem cell-related signaling and resistance to cancer chemotherapy: in fact, CSCs are known to be responsible for the therapeutic resistance and c-MYC plays a pivotal role in the regulation of CSCs [Elbadawy M, Usui T, Yamawaki H, Sasaki K. Emerging Roles of C-Myc in Cancer Stem Cell-Related Signaling and Resistance to Cancer Chemotherapy: A Potential Therapeutic Target Against Colorectal Cancer. Int J Mol Sci. 2019 May 11;20(9):2340. doi: 10.3390/ijms20092340. PMID: 31083525; PMCID: PMC6539579]. c-MYC is upregulated in several cancers and TCGA database shows that 28% of all cancers harbour genetic abnormalities in at least one of the Myc family [Schaub, F.X.; Dhankani, V.; Berger, A.C.; Trivedi, M.; Richardson, A.B.; Shaw, R.; Zhao, W.; Zhang, X.; Ventura, A.; Liu, Y.; et al. Pan-cancer alterations of the myc oncogene and its proximal network across the cancer genome atlas. Cell Syst. 2018, 6, 282–300]. c-MYC contributes to tumor progression and metastasis and induces cancer stemness properties in multiple neoplasms [Blancato, J., Singh, B., Liu, A., Liao, D. J. & Dickson, R. B. Correlation of amplification and overexpression of the c-myc oncogene in high-grade breast cancer: FISH, in situ hybridisation and immunohistochemical analyses. Br. J. Cancer 90, 1612–1619 (2004)]. It also regulates the transcription of over 15% of human genes, such as CCND, CDK4 and E2F1, which are involved in tumour progression [Meyer, N. & Penn, L. Z. Reflecting on 25 years with MYC. Nat. Rev. Cancer 8, 976–990 (2008)].  Furthermore, Strippoli et al. [Strippoli A, Cocomazzi A, Basso M, Cenci T, Ricci R, Pierconti F, Cassano A, Fiorentino V, Barone C, Bria E, Ricci-Vitiani L, Tortora G, Larocca LM, Martini M. c-MYC Expression Is a Possible Keystone in the Colorectal Cancer Resistance to EGFR Inhibitors. Cancers (Basel). 2020 Mar 10;12(3):638. doi: 10.3390/cancers12030638. PMID: 32164324; PMCID: PMC7139615] showed that alterations in the transcriptional factor c-MYC are involved in the anti-EGFR resistance in metastatic colorectal cancer: patients with higher c-MYC expression showed a significant lower PFS and OS when compared to those with low c-MYC expression. Moreover, with expression gene profiling, they pointed out the pivotal role of c-MYC in CRC-related cell-cycle, apoptosis, signal transduction and cell-growth pathways. Regarding the cell growth and proliferation pathways, they showed how UBE2C, a gene that encodes an enzyme required for cell cycle progression, is directly involved in cetuximab resistance and could be a valid target to overcome the EGFR-blockage resistance. A gene belonging to the same family (UBE2O) has been shown to promote the proliferation, epithelial–mesenchymal transformation and stemness properties of breast cancer cells through the UBE2O/AMPKα2/mTORC1-MYC positive feedback loop [Liu, X., Ma, F., Liu, C. et al. UBE2O promotes the proliferation, EMT and stemness properties of breast cancer cells through the UBE2O/AMPKα2/mTORC1-MYC positive feedback loop. Cell Death Dis 11, 10 (2020). https://doi.org/10.1038/s41419-019-2194-9]. This same gene has been demonstrated to be correlated with CBX8 levels both in colon cancer and in NAT, emphasizing the intricate interconnection between CBX8 levels, stemness, tumor progression and resistance to therapy. I think it would be useful to cite such works in order to overcome, even if only partially, the limitation linked to the absence of investigation of the functional significance of CBX8 in NAT. However, the addition of these references would guarantee a broader analysis of the topic, giving a more solid theoretical justification to the results obtained.

Response 1: Thank you very much for the comprehensive and invaluable comments. We have carefully read the comments and included these references in the discussion section of the revised manuscript.

Moreover, to show that CBX8 not only was associated with the level of CSC-associated genes, but also functionally regulated the CSC features, we performed supplementary experiments and added supplementary figure 2 in this revised manuscript. Our supplementary experiments showed that CBX8 overexpression in CRC cell-line HCT116 induced the protein level of c-MYC, which is well-known for its roles in regulating CSC feature. We further showed that CBX8 overexpression down-regulated let-7 expression, which is reported to downregulate c-MYC in CRC as well as being down-regulated by c-MYC via LIN28 repression. These results suggest that CBX8 was able to induce CSC feature through regulating c-MYC and possibly other CSC-related genes.

The following paragraph is included in the discussion section of the revised manuscript:

“In fact, CSCs are known to be responsible for the therapeutic resistance and c-MYC plays a pivotal role in the regulation of CSCs [44]. c-MYC is upregulated in several cancers and TCGA database shows that 28% of all cancers harbour genetic abnormalities in at least one of the Myc family [45]. c-MYC contributes to tumor progression and metastasis and induces cancer stemness properties in multiple neoplasms [46]. It also regulates the transcription of over 15% of human genes, such as CCND, CDK4 and E2F1, which are involved in tumour progression [47].  Furthermore, Strippoli et al. showed that alterations in the transcriptional factor c-MYC are involved in the anti-EGFR resistance in metastatic colorectal cancer: patients with higher c-MYC expression showed a significant lower PFS and OS when compared to those with low c-MYC expression [48]. Moreover, with expression gene profiling, they pointed out the pivotal role of c-MYC in CRC-related cell-cycle, apoptosis, signal transduction and cell-growth pathways. More importantly, our supplementary experiments showed that CBX8 overexpression in CRC cell-line HCT116 induced the protein level of c-MYC (supplementary figure 2), as well as repressed the level of let-7 which is reported to downregulate c-MYC in CRC as well as being down-regulated by c-MYC via LIN28 repression [49]. These results suggest that CBX8 not only was associated with the level of CSC-associated genes, but also functionally regulated the CSC features through regulating c-MYC and possibly other CSC-related genes. It is also reported that UBE2C, a gene that encodes an enzyme required for cell cycle progression, is directly involved in cetuximab resistance and could be a valid target to overcome the EGFR-blockage resistance [50]. A gene belonging to the same family (UBE2O) has been shown to promote the proliferation, epithelial–mesenchymal transformation and stemness properties of breast cancer cells through the UBE2O/AMPKα2/mTORC1-MYC positive feedback loop [51]. This same gene has been demonstrated to be correlated with CBX8 levels both in colon cancer and in NAT, emphasizing the intricate interconnection between CBX8 levels, stemness, tumor progression and resistance to therapy.

Round 2

Reviewer 1 Report

Most of the raised questions have been addressed. The manuscript is sufficiently improved.